

# The effects of resistance training to near volitional failure on motor unit recruitment during neuromuscular fatigue

Jonathan P. Beausejour[1], Kevan S. Knowles[1], Jason I. Pagan[1], Juan P. Rodriguez[1], Daniel Sheldon[1], Bradley A. Ruple[2], Daniel L. Plotkin[2], Morgan A. Smith[2], Joshua S. Godwin[2], Casey L. Sexton[2], Mason C. McIntosh[2], Nicholas J. Kontos[2], Cleiton A. Libardi[3], Kaelin Young[4], Michael D. Roberts[2] and Matt S. Stock[1]

[1] Institute of Exercise Physiology and Rehabilitation Science, University of Central Florida, Orlando, Florida, United States
[2] School of Kinesiology, Auburn University, Auburn, Alabama, United States
[3] Department of Physical Education, Federal University of Sao Carlos, Sao Carlos, Brazil
[4] Department of Biomedical Sciences, Pacific Northwest University of Health Sciences, Yakima, Washington, United States

Corresponding author
Matt S. Stock, matt.stock@ucf.edu

## ABSTRACT

**Background:** It is unclear whether chronically training close to volitional failure influences motor unit recruitment strategies during fatigue.

**Purpose:** We compared resistance training to near volitional failure *vs.* non-failure on individual motor unit action potential amplitude (MUAP) and surface electromyographic excitation (sEMG) during fatiguing contractions.

**Methods:** Nineteen resistance-trained adults (11 males, 8 females) underwent 5 weeks (3×/week) of either low repetitions-in-reserve (RIR; 0–1 RIR) or high RIR training (4–6 RIR). Before and after the intervention, participants performed isometric contractions of the knee extensors at 30% of maximal peak torque until exhaustion while vastus lateralis sEMG signals were recorded and later decomposed. MUAP and sEMG excitation for the vastus lateralis were quantified at the beginning, middle, and end of the fatigue assessment.

**Results:** Both training groups improved time-to-task failure (mean change = 43.3 s, 24.0%), with no significant differences between low and high RIR training groups (low RIR = 28.7%, high RIR = 19.4%). Our fatigue assessment revealed reduced isometric torque steadiness and increased MUAP amplitude and sEMG excitation during the fatiguing task, but these changes were consistent between groups.

**Conclusion:** Both low and high RIR training improved time-to-task failure, but resulted in comparable motor unit recruitment during fatiguing contractions. Our findings indicate that both low and high RIR training can be used to enhance fatiguability among previously resistance-trained adults.

## INTRODUCTION

Resistance training guidelines have traditionally focused on program design variables such as exercise selection, load, volume, frequency, and rest (*Haff & Triplett, 2016*). Recently, investigators have focused their attention on the extent to which a set is completed close to volitional failure (*e.g.*, whether a participant stops with repetitions in reserve [RIR]) (*Refalo et al., 2022, 2023*; *Zourdos et al., 2021*). Resistance training close to volitional failure has been proposed to enhance muscular strength, hypertrophy, and neuromuscular function in previously trained adults (*Nóbrega & Libardi, 2016*). Recently, we demonstrated that completing each set with low RIR (*e.g.*, 0–1 repetition from volitional failure) is not superior in improving muscle hypertrophy and strength outcomes when compared to high RIR (*e.g.*, 4–6 repetitions from volitional failure) in a volume-matched training program (*Ruple et al., 2023*). Although improvements in muscular strength and hypertrophy appear to be similar following low and high RIR resistance training, little is known regarding the specific neuromuscular mechanisms employed during these respective training schemes.

An explanation to address how resistance training prescription based on RIR may affect muscle strength and hypertrophic adaptations includes differences in motor unit activation and recovery. While we are unaware of previous studies that have compared the behavior of individual motor units during low *vs.* high RIR training, mechanistic insights can be derived from motor unit studies investigating fatigue during submaximal, isometric contractions (*Contessa, De Luca & Kline, 2016*; *McManus et al., 2015*; *Stock, Beck & Defreitas, 2012*). *McManus et al. (2015)* pioneered the observation of increased motor unit action potential (MUAP) amplitudes from decomposed surface electromyographic (sEMG) signals following a sustained, fatiguing contraction of the first dorsal interosseous muscle. They interpreted this as the recruitment of additional, larger motor units to maintain the desired force output. Building on this work, *Contessa, De Luca & Kline (2016)* employed a similar, but distinct fatiguing paradigm. They used repeated contractions at 30% of maximal voluntary contraction (MVC) force in the primary knee extensor muscles. Despite the differences in protocol and muscle group, *Contessa, De Luca & Kline (2016)* also demonstrated increases in sEMG excitation and MUAP amplitudes as muscles approached fatigue, corroborating and extending the findings of *McManus et al. (2015)*. Given that sets completed during low RIR training are close to volitional failure, current evidence suggests that working muscles may require compensatory strategies from the central nervous system, in part, through the additional recruitment of higher threshold motor units to maintain a given absolute work output (*Adam & De Luca, 2005*; *Contessa, De Luca & Kline, 2016*; *Contessa & Luca, 2013*). In contrast, high RIR training may not require the recruitment of additional motor units during a given set, as the included muscles are not brought close to volitional failure (*Harmon et al., 2021*; *McManus et al., 2015*). Indeed, much of the reported characteristics in motor unit recruitment during fatiguing conditions have been assessed using acute analytical designs, and motor unit characteristics following more chronic fatiguing interventions remain largely unknown.

Adaptations in motor unit activation following chronic resistance training close to volitional failure were not carefully examined until our recent work (*Ruple et al., 2023*). In

our investigation, 19 resistance-trained adults were randomly assigned to complete either low or high RIR resistance training 3×/week for five weeks. Changes in lift-specific one-repetition maximum (1RM) strength, maximal knee extensor isometric peak torque, and ultrasound-derived vastus lateralis (VL) cross-sectional area (CSA) between groups were examined. Both low and high RIR training resulted in significant improvements in 1RM strength and maximal peak torque. Slight increases in VL CSA were only found at the proximal and middle regions of the muscle, with no differences between the training groups. Although we also reported no significant group differences in maximal knee extension isometric torque following low *vs.* high RIR resistance training, changes in the mean firing rate *vs.* recruitment threshold-derived slope and y-intercept for the VL were observed among participants who completed low RIR training. Specifically, the slope decreased while the y-intercept increased, suggesting that the firing rates of earlier recruited motor units increased following chronic resistance training close to volitional failure. As changes in muscle mass and strength have been previously reported following low and high RIR training (*Refalo et al., 2022*), changes in absolute work capacity and fatiguability following these training schemes have not been adequately addressed.

Many hypotheses and discussions in the literature surrounding low *vs.* high RIR training pertain to differences in fatigue mechanisms (*Nóbrega & Libardi, 2016*). While our recent work explored motor unit adaptations specific to non-fatigued contractions (*Ruple et al., 2023*), we did not report data during fatiguing conditions. As such, the purpose of the present study was to investigate the effects of resistance training close to volitional failure on motor unit activation strategies during low torque, fatiguing contractions. Given the nature of the training interventions, we hypothesized that training close to volitional failure (*e.g.*, low RIR) would improve the time to failure more so than training further from volitional failure (*e.g.*, high RIR). Based on previous studies (*Contessa, De Luca & Kline, 2016*; *Harmon et al., 2021*; *Mota et al., 2020*), we further speculated that sEMG excitation and MUAP would increase throughout the fatiguing task, with similar changes between groups.

## MATERIALS AND METHODS

### Research design

The data presented herein was collected as a secondary aim from our previous study in which the experimental procedures were reviewed and approved by the Auburn University Institutional Review Board (IRB approval #: 21-507 MR 2111). Experimental procedures were conducted according to the standards set by the latest revisions of the Declaration of Helsinki, except for the study being registered as a clinical trial. Resistance-trained males and females were randomly assigned to low or high RIR training. Before and after the intervention, the participants visited the laboratory for data collection. During each data collection session, participants completed isometric MVCs of their knee extensors, followed by repeated submaximal contractions to task failure at 30% MVC torque. sEMG sensors were placed on the VL to record electrical activity of the muscle. Participants were asked to refrain from physically demanding activity for ≥24 h before each visit.

## Research participants

Eleven resistance-trained males (mean ± SD age = 24 ± 3 years, mass = 91.5 ± 22.5 kg, height = 180.7 ± 6.9 cm) and eight resistance-trained females (age = 25 ± 2 years, mass = 61.9 ± 4.8 kg, height = 163.4 ± 6.4 cm) completed this study and were randomly assigned to complete low (*n* = 10; six males, four females) or high (*n* = 9; five males, four females) RIR resistance training. Each participant reported consistent resistance training experience over the previous year. Resistance training status was further verified *via* 1 RM screening, in which male participants were able to squat >1.5× and females >1.15× body weight. Participants were screened to ensure the absence of cardiometabolic/neuromuscular diseases, refrainment from consuming hormone-altering substances within the previous 2 months (excluding invasive/oral contraceptives), and detailed disclosure of prescription and over-the-counter medication. All participants were asked to refrain from additional resistance training beyond the study and maintain their current nutritional practices. Each participant read, understood, and signed an informed consent document prior to study enrollment.

## Isometric torque fatiguing task

Muscle strength testing was performed with a Biodex System 4 isokinetic dynamometer (Biodex Medical Systems, Shirley, NY, USA) for isometric torque assessment of the participants' dominant knee extensors (based on kicking preference). Participants were securely restrained on the dynamometer chair with straps around both shoulders, hips, and thigh. The dynamometer seat was adjusted so that the participants' dominant knee joint was aligned with the axis of the dynamometer lever arm and fixed at 90°. The hip joint was also positioned at 90°. Prior to testing, participants performed a warm-up set that consisted of three, 10-s, isometric contractions corresponding to 50%, 70%, and 90% of their perceived maximum. Following the warm-up, participants performed 3, 3-s MVCs, with each trial separated by ≥3 min of rest (*Contessa, De Luca & Kline, 2016*; *Harmon et al., 2021*). The highest peak torque value from the three trials was designated as the MVC torque value (Nm) and used for the subsequent fatiguing task. Following MVC determination, the participants performed an isometric fatigue protocol, which required them to complete trapezoidal isometric contractions of the knee extensors at 30% MVC until exhaustion. For each contraction, knee extension torque was increased from 0 to 30% in 3 s (10.0%/s), held constant for 33 s, and decreased from 30 to 0% in 3 s (10.0%/s) (*Harmon et al., 2021*). To allow for brief periods of no muscle activity, 3 s were allotted before and after each contraction, thereby providing 6 s of rest between contractions. Participants repeated this 30% MVC series for as many cycles as possible. Repeated cycles at 30% MVC were administered to slowly induce fatigue, as 30% MVC has been identified as an appropriate low intensity level to induce gradual muscle fatigue during recurrent isometric contractions in the knee extensor muscles (*Pethick & Tallent, 2022*). Time to task failure (seconds) was quantified by the completed duration of the fatiguing protocol (including rest between contractions) and terminated when isometric torque dropped below 25% MVC for two consecutive seconds. The absolute torque value that was utilized during baseline testing was also used during post-intervention testing, specifically, to

evaluate performance changes in absolute load intensities. In addition, torque steadiness was quantified as the coefficient of variation (%) of the torque signal for a 2-s epoch at the normalized beginning, middle, and end of each participant's constant torque time.

## sEMG signal recording

Bipolar sEMG signals were recorded from the VL muscle with a Bagnoli 16-channel desktop system (Delsys, Inc., Natick, MA, USA). Prior to sEMG signal acquisition, the skin over the VL muscle and ipsilateral patella was shaved and cleansed with rubbing alcohol. Dead skin cells, debris, and oil were removed with hypoallergenic tape. A reference electrode was positioned over the patella. The sensor utilized for motor unit analysis was placed over the VL muscle, in accordance with the specific landmark parameters outlined by *Zaheer, Roy & De Luca (2012)*. The bipolar signals were detected with a 5 × 5 mm sEMG sensor (Delsys, Inc., Natick, MA, USA) that consists of five pin electrodes (*Nawab, Chang & De Luca, 2010*). Four of the five electrodes are arranged in a square, with the fifth electrode in the center of the square and at an equal distance from all other electrodes (3.6 mm). Pairwise subtraction of the five electrodes was used to derive four single differential sEMG channels. These signals were differentially amplified and filtered with bandwidth of 20 to 450 Hz and sampled at 20 kHz. Prior to data acquisition, a sEMG signal-to-noise ratio >3.0 and line interference <1.0 were all ensured during a 20% MVC torque sEMG signal quality check.

## sEMG excitation

sEMG excitation was quantified as the mean of the root mean square values (plural) across the four separate filtered sEMG signals obtained from the array sensor. The same 2-s beginning, middle, and end epochs previously used to quantify torque steadiness were utilized for analyses of sEMG excitation. In situations where the fatigue protocol ended prior to the completion of the final contraction, the last 2 s of the plateau were analyzed, and the epoch selection for the middle 2 s was normalized for total time (*Harmon et al., 2021*).

## sEMG decomposition and MUAP amplitude

Following data acquisition, the four separate filtered sEMG signals from the VL were decomposed into their constituent MUAP shapes and trains *via* the Precision Decomposition III (PD III) algorithm (*De Luca et al., 2006*; *Nawab, Chang & De Luca, 2010*). Once the signals were successfully decomposed, the reconstruct-and-test procedure was used to determine the accuracy of each identified motor unit (*De Luca & Contessa, 2012*; *Nawab, Chang & De Luca, 2010*). MUAP trains identified at the ≥90% accuracy threshold level were included for further analysis (*Herda et al., 2020*).

A custom LabVIEW program (version 21.0, National Instruments, Austin, TX, USA) was utilized to quantify the MUAP peak-to-peak amplitude from each of the four channels. The mean peak-to-peak amplitude value across the four channels was used for all subsequent analyses (*Harmon et al., 2021*; *Pope et al., 2016*). MUAP amplitude was quantified for each motor unit at the beginning, middle, and end of the fatigue protocol.

The first and last contraction performed by each participant during the fatigue protocol was considered the "beginning" and "end" fatigue, respectively. The "middle" fatigue was considered the middle contraction. In situations when an even number of contractions was performed, middle fatigue was considered the earlier of the contractions. For example, if a participant successfully completed six contractions, MUAP amplitude data were analyzed for the first contraction (beginning), third contraction (middle), and sixth contraction (end). Furthermore, MUAP values were averaged and plotted across each performed contraction to analyze changes in MUAP values throughout the fatigue protocol (*Harmon et al., 2021*; *Mota et al., 2020*). MUAP changes across the fatiguing protocol were characterized by the slope value of the linear-regressed relationship from the mean MUAP values computed for each contraction (mV/contraction). For instance, if a participant completed four contractions during the fatigue protocol, linear regression variables (*e.g.*, slope) were derived from the four computed mean MUAP values. An example of MUAP data from one participant who performed four contractions is displayed in Fig. 1, and visualization of fatigue paradigm may be found in *Harmon et al. (2021)*.

### Resistance training program

Following the isometric torque fatiguing taks, participants were randomly assigned to complete low or high RIR training for 5 weeks. Group assignment was based on the participants' Wilks score (as determined from one-repetition maximum testing), which normalizes strength across different bodyweights and sex (*Vanderburgh & Batterham, 1999*). The mean Wilks scores were not different between the two groups (Low RIR: 269.3 ± 46.0, High RIR: 263.0 ± 55.4, *p* = 0.79). Following group assignment, all participants completed a similar full body training program, three times per week, for 5 weeks, followed by an additional week of de-load exercises of barbell squat, barbell bench press, and barbell deadlift exercises at reduced training volumes. The resistance training program was progressively overloaded each week, with resistance intensities ranging from 65–95% 1 RM. During the 5-week training period, the low RIR group was directed to complete each set of the back squat, bench press, and deadlift exercises as close to volitional failure as possible (*e.g.*, RIR of 0–1). The high RIR group was directed to complete each set of the squat, bench press, and deadlift exercises at the pre-determined training prescription, adjusting the load appropriately to not reach volitional failure (*e.g.*, RIR of 4–6). The main exercises (*e.g.*, squat, bench press, deadlift) were the only exercises performed to near volitional failure by the low RIR group (Table S1). The RIR approach implemented in the study was in accordance with parameters outlined by *Helms et al. (2016)* and *Zourdos et al. (2016)*. There was no direct supervision of the participants' resistance training sessions. Participants completed the training sessions at local gymnasiums, and logged training details on a shared file accessible to only the participant and study coordinators. Sheets were inspected thoroughly daily, and participants were contacted *via* phone or email to ensure that training sessions were completed correctly (according to group assignment), and training logs were updated promptly.

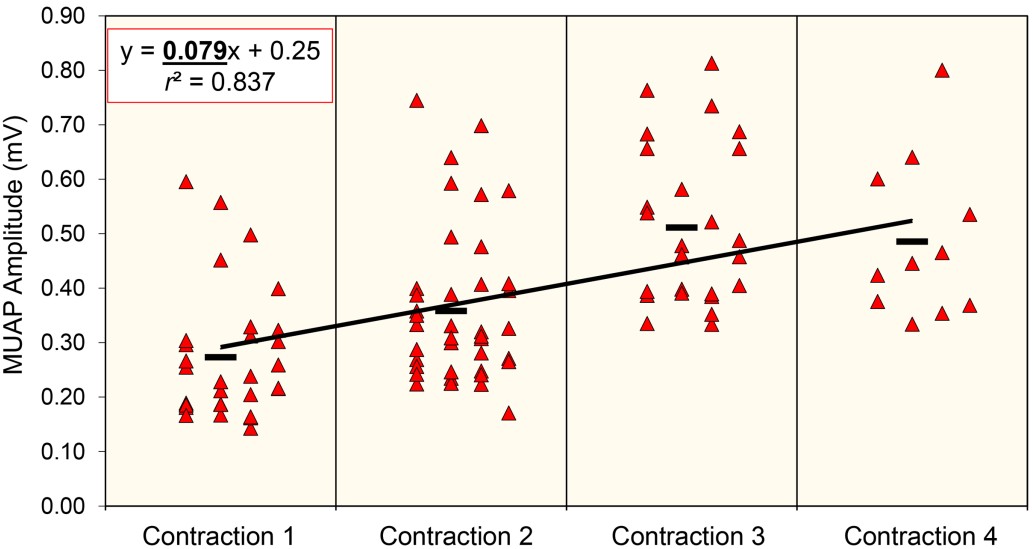

**Figure 1 Exploration of motor unit action potential amplitude during fatigue.** Example data of the MUAP slope for one participant. Each data point represents the MUAP amplitude value (mV) for identified motor units within each contraction completed during the fatiguing protocol. The black bars represent the mean MUAP value for the contraction. The linear regression line has been derived from the mean MUAP values across the four contractions.

## Statistical analyses

We explored mean differences *via* Mann-Whitney U tests and mixed factorial (*e.g.*, time × group) analyses of variance (ANOVAs). For analyses with two factors (time to task failure and mean MUAP slope), two-way mixed factorial ANOVAs were utilized. Torque steadiness, sEMG excitation, and mean MUAP amplitude were examined with separate three-way (time (pre *vs.* post) × fatigue time interval (beginning *vs.* middle *vs.* end) × group (low RIR *vs.* high RIR) ANOVAs. For factors with three levels, Greenhouse-Geisser corrections were applied when the sphericity assumption was violated. Significant interactions or main effects were evaluated with Bonferroni-corrected pairwise comparisons and Cohen's d effect sizes. For the pairwise comparisons, effect sizes were evaluated with 0.2, 0.5, and 0.8 corresponding to small, medium, and large effect sizes, respectively. Effect sizes for the Mann-Whitney U tests (r) were evaluated with 0.1, 0.3, and 0.5 corresponding to small, medium, and large effect sizes, respectfully. Partial eta squared values were evaluated as measures of effect size for each ANOVA, with 0.01, 0.06, and 0.14 corresponding to small, medium, and large effect sizes, respectively (*Cohen, 1988*). An alpha level of $p < 0.05$ was used for all statistical analyses. All statistical analyses were conducted with JASP software (version 0.16.3; JASP Team, University of Amsterdam, Amsterdam, Noord-Holland).

## RESULTS

Table 1 presents baseline characteristics for time to task failure and mean MUAP slope for each group prior to training period. Tables 2 and 3 presents ANOVA results for the
**Table 1 Baseline data.**

| | Low RIR | High RIR | p-value | Effect size (r) |
|---|---|---|---|---|
| Time to task failure (seconds) | 112.4 ± 56.5 | 162.3 ± 67.6 | 0.095 | 0.467 |
| MUAP slope (mV/contraction) | 0.035 ± 0.04 | 0.027 ± 0.02 | 0.930 | 0.037 |

**Note:**
Mean ± SD baseline data for time to task failure performance (seconds) and the mean MUAP slope (mV/contraction). Mann-Whitney U statistics and effect sizes (r) were used to compare baseline measures.

**Table 2 2-way ANOVA results.**

| | Group × Time | Group [low RIR vs. high RIR] | Time [pre-test vs. post-test] |
|---|---|---|---|
| **Time to task failure** | $F = 0.180$ | $F = 2.7$ | *$F = 19.3$ |
| | $p = 0.677$ | $p = 0.118$ | $p < 0.001$ |
| | $\eta p^2 = 0.010$ | $\eta p^2 = 0.139$ | $\eta p^2 = 0.531$ |
| **Mean MUAP slope** | $F = 0.265$ | $F = 0.955$ | $F = 3.3$ |
| | $p = 0.614$ | $p = 0.343$ | $p = 0.090$ |
| | $\eta p^2 = 0.016$ | $\eta p^2 = 0.056$ | $\eta p^2 = 0.169$ |

**Notes:**
ANOVA results for two-way (group (low RIR *vs.* high RIR)) × time (pre *vs.* post) × time interval (beginning *vs.* middle *vs.* end)) interactions for isometric peak torque, time to task failure, and mean MUAP slope.
* Bold = significant observation.

**Table 3 3-way ANOVA results.**

| | Group × Time × Fatigue time interval | Group × Time | Group × Fatigue time interval | Time × Fatigue time interval | Group | Time | Fatigue time interval |
|---|---|---|---|---|---|---|---|
| **Torque steadiness** | $F = 1.899$ | $F = 0.058$ | $F = 0.357$ | $F = 0.213$ | $F = 0.428$ | $F = 0.31$ | *$F = 14.7$ |
| | $p = 0.177$ | $p = 0.813$ | $p = 0.639$ | $p = 0.746$ | $p = 0.522$ | $p = 0.582$ | $p < 0.001$ |
| | $\eta p^2 = 0.100$ | $\eta p^2 = 0.003$ | $\eta p^2 = 0.021$ | $\eta p^2 = 0.012$ | $\eta p^2 = 0.025$ | $\eta p^2 = 0.018$ | $\eta p^2 = 0.464$ |
| **sEMG excitation** | $F = 0.089$ | $F = 0.389$ | $F = 0.270$ | $F = 1.477$ | $F = 4.24^{10 \times -4}$ | $F = 1.11$ | *$F = 13.8$ |
| | $p = 0.822$ | $p = 0.542$ | $p = 0.636$ | $p = 0.245$ | $p = 0.984$ | $p = 0.304$ | $p < 0.001$ |
| | $\eta p^2 = 0.006$ | $\eta p^2 = 0.024$ | $\eta p^2 = 0.017$ | $\eta p^2 = 0.085$ | $\eta p^2 = 2.65^{10 \times -5}$ | $\eta p^2 = 0.066$ | $\eta p^2 = 0.464$ |
| **Mean MUAP amplitude** | $F = 0.740$ | $F = 0.209$ | $F = 0.463$ | $F = 0.418$ | $F = 0.037$ | $F = 1.50$ | *$F = 15.9$ |
| | $p = 0.827$ | $p = 0.654$ | $p = 0.517$ | $p = 0.558$ | $p = 0.850$ | $p = 0.238$ | $p < 0.001$ |
| | $\eta p^2 = 0.005$ | $\eta p^2 = 0.013$ | $\eta p^2 = 0.028$ | $\eta p^2 = 0.025$ | $\eta p^2 = 0.002$ | $\eta p^2 = 0.086$ | $\eta p^2 = 0.498$ |

**Notes:**
ANOVA results for three-way (group (low RIR *vs.* high RIR)) × time (pre *vs.* post) × fatigue time interval (beginning *vs.* middle *vs.* end)) and subsequent two-way interactions for torque steadiness, sEMG excitation, and mean MUAP amplitude.
* Bold = significant observations.

two-factor (time to task failure and mean MUAP slope) and three-factor (torque steadiness, sEMG excitation, and mean MUAP amplitude) analyses, respectively. Tables 4 and 5 presents the computed means, standard errors, and 95% confidence intervals for the two-factor and three-factor analyses, respectfully.

**Table 4  Mean MUAP slope.**

|  | Group | | Time | |
|---|---|---|---|---|
|  | Low RIR | High RIR | PRE | POST |
| Time to task failure (s) | 136.10 ± 19.13 | 181.90 ± 20.16 | 137.34 ± 14.23 | 180.65 ± 15.24 |
|  | (95.74–176.46) | (139.36–224.44) | (107.31–167.37) | (148.49–121.82) |
| Mean MUAP Slope (mV/contraction) | 0.032 ± 0.008 | 0.021 ± 0.008 | 0.031 ± 0.008 | 0.021 ± 0.004 |
|  | (0.015–0.048) | (0.004–0.037) | (0.015–0.047) | (0.012–0.031) |

Note:
Bold text = Mean ± standard error (95% confidence intervals) values for time to task failure and mean MUAP slope.

**Table 5  Torque steadiness, sEMG Excitation, MUAP amplitude.**

|  | Group | | Time | | Fatigue time interval | | |
|---|---|---|---|---|---|---|---|
|  | Low RIR | High RIR | PRE | POST | Beginning | Middle | End |
| Torque steadiness (%) | 3.33 ± 0.33 | 3.64 ± 0.35 | 3.60 ± 0.24 | 3.38 ± 0.37 | 2.47 ± 0.26 | 2.89 ± 0.22 | 5.10 ± 0.57 |
|  | (2.63–4.03) | (2.91–4.38) | (3.09–4.10) | (2.61–4.15) | (1.92–3.03) | (2.42–3.36) | (3.90–6.30) |
| sEMG excitation (RMS μV) | 83.30 ± 9.96 | 83.04 ± 9.96 | 88.00 ± 9.70 | 78.37 ± 6.80 | 63.00 ± 2.93 | 85.57 ± 7.37 | 100.98 ± 0.57 |
|  | (62.23–104.43) | (61.94–104.14) | (67.44–108.55) | (63.98–92.77) | (56.78–69.22) | (69.96–101.18) | (76.03– 125.93) |
| Mean MUAP amplitude (mV) | 0.20 ± 0.04 | 0.19 ± 0.04 | 0.21 ± 0.03 | 0.18 ± 0.03 | 0.14 ± 0.02 | 0.21 ± 0.03 | 0.23 ± 0.04 |
|  | (0.12–0.28) | (0.11–0.27) | (0.14–0.28) | (0.13–0.23) | (0.10–0.17) | (0.15–0.27) | (0.16–0.31) |

Note:
Bold text = Mean ± standard error (95% confidence intervals) values for torque steadiness, sEMG excitation, and mean MUAP amplitude.

## Baseline measures

Results from the Mann-Whitney's U tests indicated no baseline differences between groups for time to task failure ($p = 0.095$) and MUAP slope ($p = 0.930$). A large effect size was observed for time to task failure, indicating that the high RIR group achieved a longer time to task failure compared to low RIR group at baseline (mean difference = 30.8%).

## Time to task failure

The results from the two-way mixed factorial ANOVA indicated no time × group interaction (Fig. 2). However, there was a significant main effect for time, suggesting that all participants showed improvement in time to task failure (total mean change = 43.3 s, 24.0%). Mean changes following the intervention period were slightly larger for the low RIR group compared to the high RIR group (pre to post intervention mean ± standard deviation for low RIR group: 112.4 ± 56.5 to 159.9 ± 48.4 s [28.7% longer]; high RIR group: 162.3 ± 67.6 to 201.5 ± 82.0 s [19.4% longer]).

## Torque steadiness

The results from the three-way mixed factorial ANOVAs indicated that there were no significant interactions with small to medium effect sizes for all comparisons ($F \leq 1.899$, $p \geq 0.177$, $\eta p^2 \leq 0.100$). However, there was a significant main effect for the fatigue time interval (Fig. 3A). When collapsed across group and time, torque steadiness was significantly greater (e.g., higher coefficient of variation, $p < 0.001$, d ≥ 1.000) at the end of

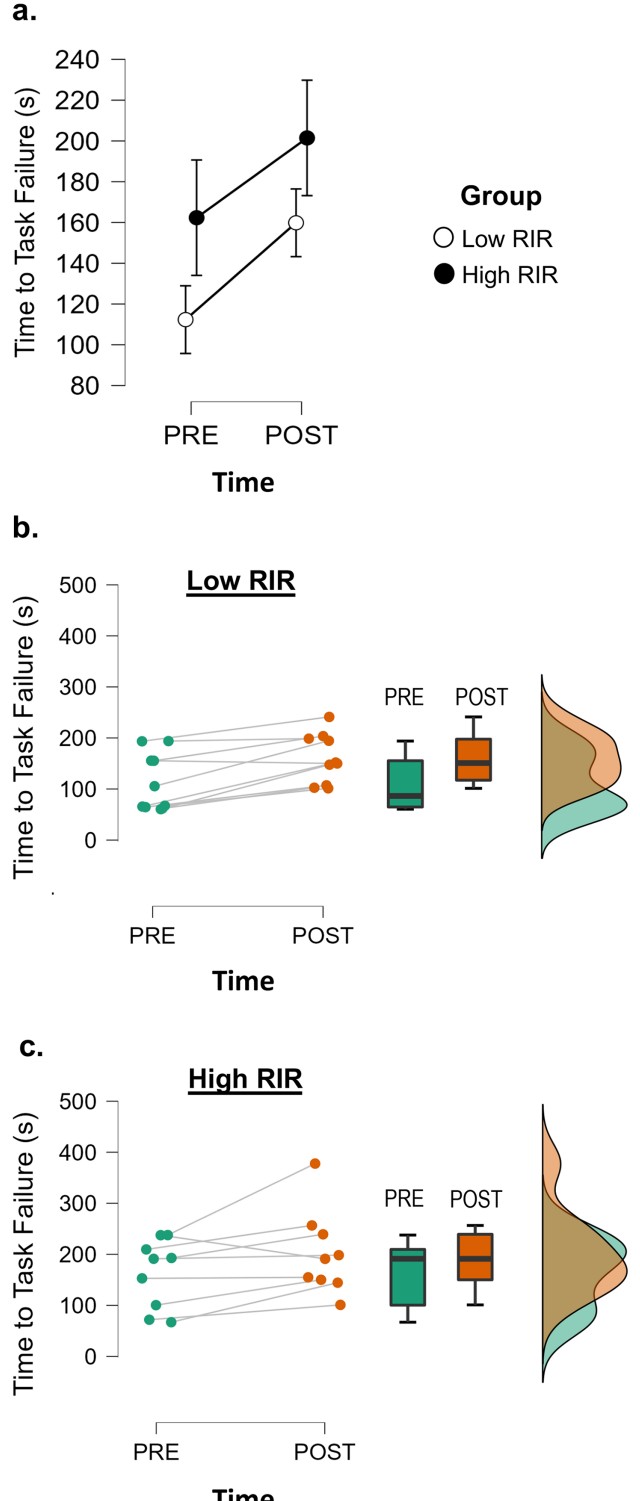

**Figure 2 Both low and high RIR training enhance fatigability.** JASP ANOVA and raincloud plots displaying between group (A) and individual differences (B and C) in time to task failure (seconds) before and after the training intervention. The middle plot (B) displays PRE-POST changes for each participant in the low RIR group. The bottom plot (C) displays PRE-POST changes for each participant in the high RIR group.

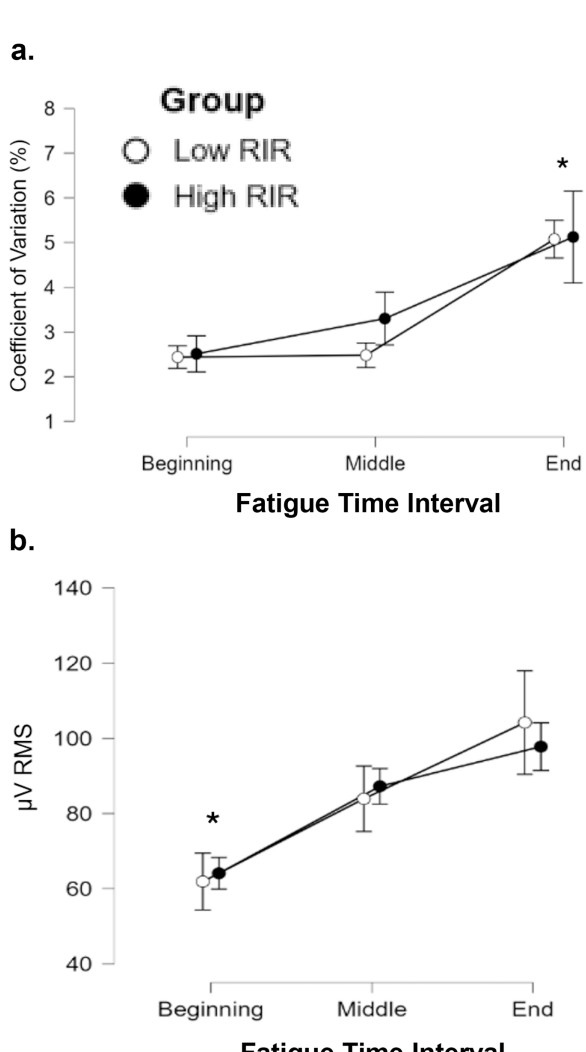

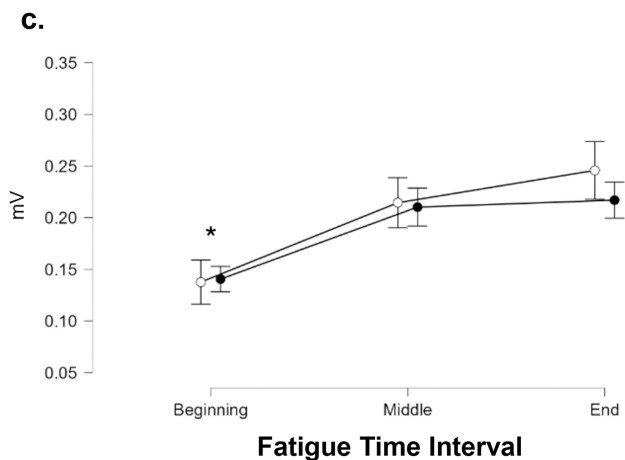

**Figure 3 Isometric fatigue induces reductions in torque steadiness and increases in sEMG and MUAP amplitude, independent of low and RIR training.** A JASP descriptive plot displaying time-collapsed changes in torque steadiness (A), sEMG excitation (B), and mean MUAP amplitude (C) across fatigue time intervals for each group. An asterisk (*) indicates significant observation for the specific fatigue time interval.

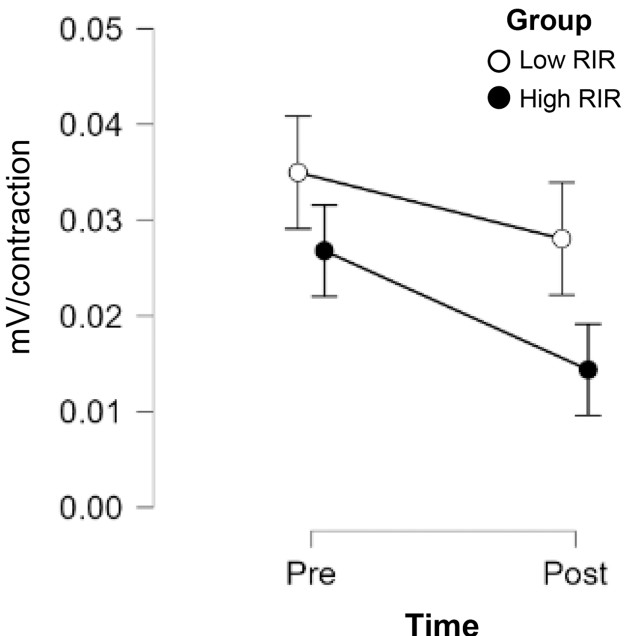

**Figure 4 Non-significant declines in MUAP slope following low and high RIR training.** A JASP ANOVA plot displaying PRE-POST changes in the mean MUAP slope between groups.

the isometric fatigue task (beginning = middle < end; mean ± standard error = 2.74 ± 0.26, 2.89 ± 0.22, 5.09 ± 0.57%, respectively).

### sEMG excitation

The results from the three-way mixed factorial ANOVAs indicated that there were no significant interactions with small to medium effect sizes for all comparisons ($F \leq 1.477$, $p \geq 0.245$, $\eta p^2 \leq 0.085$). There was a significant main effect for fatigue time interval (Fig. 3B). When collapsed across group and time, the sEMG excitation was smaller during the beginning of the isometric fatigue task ($p \leq 0.012$, d ≥ 0.547). No differences were observed between the middle and end intervals (beginning < middle = end; mean ± standard error = 63.0 ± 8.2, 85.6 ± 8.2, 101.0 ± 8.1 µV RMS, respectively).

### Mean MUAP amplitude

The results from the three-way mixed factorial ANOVAs indicated that there were no significant interactions with small to medium effect sizes for all comparisons ($F \leq 0.463$, $p \geq 0.517$, $\eta p^2 \leq 0.028$). There were no significant main effects for time or group. There was, however, a significant main effect for fatigue time interval (Fig. 3C). When collapsed across group and time, mean MUAP amplitude was smaller during the beginning of the isometric fatigue task ($p < 0.001$, d ≥ 0.553). No differences were observed between the middle and end intervals (beginning < middle = end; mean ± standard error = 0.139 ± 0.028, 0.212 ± 0.029, 0.231 ± 0.029 mV, respectively).
### Mean MUAP amplitude slope

The results from the two-way mixed factorial ANOVA indicated no time × group interaction, and no main effect for group or time (Fig. 4). However, a large effect size was observed for time ($\eta p^2 = 0.169$; pre to post intervention mean ± standard deviation for Low RIR group: 0.035 ± 0.040 to 0.028 ± 0.024 mV/contraction; high RIR group: 0.027 ± 0.022 to 0.014 ± 0.012 mV/contraction).

## DISCUSSION

Our novel study was the first to compare changes in time to task failure and motor unit control following chronic resistance training to near volitional failure (low-RIR) or non-failure (high-RIR). Given that sets completed during low RIR training are performed closer to volitional exhaustion, we hypothesized that participants assigned to the low RIR group would show greater improvements in time to task failure than those performing high RIR resistance training. Contrary to our hypothesis, the improvements in time to task failure were not different between groups. Although we observed significant declines in torque steadiness and significant increases in MUAP amplitude and sEMG excitation during the fatiguing task, there were no differences between the groups following training. Below we discuss the interpretation, strengths, and limitations of our work, in addition to areas for further inquiry.

Previous investigations have reported similar improvements in lower body specific 1 RM strength (*Nóbrega & Libardi, 2016*; *Santanielo et al., 2020*) and knee extensor isometric MVC torque (*Fisher, Blossom & Steele, 2016*; *Lacerda et al., 2020*) following low and high RIR training. Combined with the similar improvements in isometric peak torque and 1RM performance between the training groups in our previous study (*Ruple et al., 2023*), the improvements in time to task failure were also not different between groups. As such, our results indicate that improvement in absolute work capacity may be mediated by increases in muscular strength following training, regardless of whether the resistance training sets are completed close to failure. For reference, *Izquierdo et al. (2009)* reported a 10.8% increase in isometric knee extensor strength and 15.5% increase in lower body work capacity following 7 weeks of heavy resistance training in physically active men. Our previously reported improvement in isometric knee torque following the training period among all included participants (10.1%) resembles the increase reported in the *Izquierdo et al. (2009)* investigation, with the accompanying 24.0% improvement in the time to failure task. *Byrd et al. (2021)* further demonstrated that 1RM back squat strength significantly contributed to the prediction of anaerobic work capacity (derived *via* 3-min cycle ergometer critical power test) in anaerobically trained participants. The relative improvement in isometric peak torque, 1RM, and time to task failure in our investigations appear to be consistent regardless of low or high RIR training schemes. Considering the novelty of the present study, additional research is warranted to further parse out the relationship between acquired strength gains and relative improvements in task-specific work capacity following low *vs.* high RIR resistance training.

Decreases in torque steadiness during repeated submaximal isometric contractions to muscle fatigue are well established within the literature (*Castronovo et al., 2015*; *Jones et al.,*

*2023*; *Pethick & Tallent, 2022*). These characteristics were also observed in the present study, as torque steadiness decreased at a comparable rate during the fatiguing task in both training groups (~42.3% between the middle and end fatigue time intervals), and may be explained through altered neural drive to the working muscle *via* changes in common motor neuron synaptic input (*Enoka & Farina, 2021*; *Pethick & Tallent, 2022*). sEMG excitation responses to the fatiguing task were also not different between training groups and confirm previously reported characteristics of sEMG excitation during isometric, fatiguing protocols (*Harmon et al., 2021*; *Miller et al., 2020*; *Muddle et al., 2018*). Specifically, sEMG excitation increased at a comparable rate between the low and high RIR groups, which further explains the null group differences in total motor unit recruitment during muscle fatiguing conditions (*e.g.*, similar trends in input excitation during the fatiguing task). As previous literature presented evidence for the affinity of the central nervous system to recruit additional higher-threshold motor units as an acute compensation strategy during fatigue, relevant investigations have also suggested that additional motor unit recruitment will occur sooner if the muscle is in a fatigued state (*Contessa, De Luca & Kline, 2016*; *McManus et al., 2015*). Until the present investigation, it was unclear whether the rate of this additional motor unit recruitment will be altered following chronic training to near volitional failure. Our results, however, suggest that the rate of additional motor unit recruitment may be primarily dependent upon the fatigue status of the muscle and not an adaptive response to chronic, low RIR training. Of relevance, our reported characteristics of sEMG excitation and motor unit recruitment following training were observed at the same absolute intensity as implemented prior to the training period (*i.e.*, the fatiguing task was conducted at the same absolute torque level across testing visits). Administering the fatiguing protocol while also accounting for the strength gains achieved by the included participants may yield different motor unit activation trends than presently observed.

An interesting observation reported in our previous study was the significant change in slope and y-intercepts of the mean firing rate *vs.* recruitment threshold relationship that completed low RIR training. As these outcomes were obtained *via* pre-post sEMG decomposition analysis of isometric trapezoidal contractions at 80% MVC torque level, the slope values decreased while the y-intercept values increased, suggesting that the firing rates of earlier-recruited motor units increased following chronic resistance training close to volitional failure. This was noted despite no significant group differences in the improvement of knee extension isometric torque, although a medium to large effect size was observed for the low RIR group. The medium-to-large effect size observed in isometric torque were also coupled with the large effect sizes observed in total training volume for the low RIR group, in which notable (but insignificant) differences in training volume were observed between groups during the first 3 weeks of the training period. This trend suggests that the disparity in training volume and RIR was greater during the first few weeks of training (*i.e.*, weeks 1–3) compared to the latter weeks (*i.e.*, weeks 4 and 5), despite participants in the high RIR group prioritizing RIR around 2–3 for relatively higher training loads. Given that the training intervention prescribed lower loads during the earlier weeks of training, an argument can be presented that the mechanisms which

ultimately led to group differences in motor unit characteristics may be explained through the first few weeks of the training program. The low RIR group was observed to have higher firing rates of earlier-recruited motor units, which may implicate specific motor unit adaptations when relatively low loads are performed close to volitional failure. Specifically, each set completed during low RIR training may further exhaust the functional capacity of lower threshold motor units (*i.e.*, increase their firing rates) while higher threshold motor units are subsequently recruited. The increase in motor unit firing rates observed in the low RIR group, however, did not lead to different motor unit recruitment trends in the present study, as no significant group differences were observed in sEMG excitation or MUAP trends during the fatiguing task. This may be relevant, as VL sEMG excitation and MUAP trends were reported to be similar between the 80% MVC task and the end time interval of the fatiguing task administered in our collective investigations (*Harmon et al., 2021*). It is also important to note that changes in motor unit firing rates following resistance training, specifically, remain quite mixed (*Herda, 2022*). Future investigations are needed to appropriately characterize motor unit recruitment and firing rate modulation during fatigue following low and high RIR training schemes.

This study has multiple limitations that should be considered. First, our resistance training intervention was administered without direct supervision, and it is possible that the study participants were not truthful or made errors in logging the details of their training sessions. To account for participant training adherence and accountability, frequent check-ins with each participant were utilized. Second, the prescribed progressive overload in our resistance training intervention gradually reduced the RIR disparity between groups, particularly in the later weeks. This diminishing RIR gap may have affected the effectiveness of our between-group intervention and added complexity to the interpretation of our analytical comparisons. While this convergence of RIR values in the latter stages introduces some nuance to our findings, it also reflects the real-world application of progressive overload principles in resistance training. Third, the duration of training intervention (5 weeks of progressive overload and 1-week deload) may have been too short to result in motor unit adaptations among previously resistance-trained participants, which may have had implications for our MUAP interpretation during the fatiguing task (*Herda, 2022*). Relatedly, while the training program did involve exercises that relied on knee extension torque, the fatiguing task used to study motor unit adaptations did not directly reflect the participants' training, and thus it is possible that the lack of task specificity may have influenced our results. Although no significant differences were observed in our dependent variables at baseline, the large effect size observed between groups for time to task failure may have influenced our interpretation. resistance training program. Our study also reported data for 19 participants, which may have limited our statistical power. As such, we also utilized effect sizes in the interpretation of our data. Furthermore, the eight females that participated in this study were enrolled without considering specific details regarding their menstrual cycle phase. The literature regarding motor unit adaptations during fatigue in females in different menstrual cycle phases warrant future attention (*Lulic-Kuryllo & Inglis, 2022*). Importantly, we aimed to characterize adaptations in groups of motor units, based on the amplitude of individual

motor units and in accordance with previous investigations that utilized similar methods (*Contessa et al., 2018*; *Harmon et al., 2021*; *Mota et al., 2020*; *Muddle et al., 2018*). Given the complexities associated with analyzing motor unit data during fatigue *via* sEMG decomposition algorithms (*Dimitrova & Dimitrov, 2003*; *Nawab, Chang & De Luca, 2010*), we cannot fully guarantee that the same motor units were recorded across the included isometric contractions across lab visits. Taken together, these strengths and limitations should be considered when reviewing our study findings in the context of the broader RIR resistance training literature.

## CONCLUSIONS

In conclusion, the results of the present study provide evidence that moderate to high intensity resistance training (*e.g.*, 65–95% 1RM) using low (0–1) *vs.* high (4–6) RIR results in similar improvements in time to task failure during submaximal, isometric contractions. We also observed significant declines in torque steadiness and the recruitment of additional high-threshold motor units during a submaximal isometric fatiguing task, but there were no unique adaptations over time or between groups. As such, high RIR training appears to be as effective as low RIR training in improving muscular work capacity. With respect to low RIR training schemes, the task-specific nature of chronic training to near volitional failure did not provide any additive benefit compared to high RIR training. Our findings add to a growing body of literature showing similar performance benefits following low *vs.* high RIR training.

## ACKNOWLEDGEMENTS

We extend our sincerest appreciation to the individuals who volunteered to participate in this study.

### Funding

Mason C McIntosh was fully supported through a T32 NIH grant (T32GM141739). Cleiton A Libardi was supported by The São Paulo Research Foundation (n° 2020/13613-4) and National Council for Scientific and Technological Development (CNPq) (n° 311387/2021-7). The funders had no role in study design, data collection and analysis, decision to publish, or preparation of the manuscript.

### Grant Disclosures

The following grant information was disclosed by the authors:
T32 NIH: T32GM141739.
The São Paulo Research Foundation: n° 2020/13613-4.
National Council for Scientific and Technological Development: (CNPq) n° 311387/2021-7.

### Competing Interests

The authors declare that they have no competing interests.

## Author Contributions

- Jonathan P. Beausejour conceived and designed the experiments, performed the experiments, analyzed the data, prepared figures and/or tables, authored or reviewed drafts of the article, and approved the final draft.
- Kevan S. Knowles analyzed the data, prepared figures and/or tables, authored or reviewed drafts of the article, and approved the final draft.
- Jason I. Pagan performed the experiments, prepared figures and/or tables, authored or reviewed drafts of the article, and approved the final draft.
- Juan P. Rodriguez performed the experiments, authored or reviewed drafts of the article, and approved the final draft.
- Daniel Sheldon performed the experiments, authored or reviewed drafts of the article, and approved the final draft.
- Bradley A. Ruple conceived and designed the experiments, performed the experiments, authored or reviewed drafts of the article, and approved the final draft.
- Daniel L. Plotkin performed the experiments, authored or reviewed drafts of the article, and approved the final draft.
- Morgan A. Smith performed the experiments, authored or reviewed drafts of the article, and approved the final draft.
- Joshua S. Godwin performed the experiments, authored or reviewed drafts of the article, and approved the final draft.
- Casey L. Sexton performed the experiments, authored or reviewed drafts of the article, and approved the final draft.
- Mason C. McIntosh performed the experiments, authored or reviewed drafts of the article, and approved the final draft.
- Nicholas J. Kontos performed the experiments, authored or reviewed drafts of the article, and approved the final draft.
- Cleiton A. Libardi performed the experiments, authored or reviewed drafts of the article, and approved the final draft.
- Kaelin Young performed the experiments, authored or reviewed drafts of the article, and approved the final draft.
- Michael D. Roberts conceived and designed the experiments, performed the experiments, authored or reviewed drafts of the article, and approved the final draft.
- Matt S. Stock conceived and designed the experiments, performed the experiments, analyzed the data, prepared figures and/or tables, authored or reviewed drafts of the article, and approved the final draft.

## Human Ethics

The following information was supplied relating to ethical approvals (*i.e.*, approving body and any reference numbers):

The experimental procedures were reviewed and approved by the Auburn University Institutional Review Board who, granted approval to carry out the study within its facilities according to the latest revisions of the Declaration of Helsinki (IRB approval #: 21-507 MR 2111).

## Ethics

The following information was supplied relating to ethical approvals (*i.e.*, approving body and any reference numbers):

The experimental procedures were reviewed and approved by the Auburn University Institutional Review Board (IRB approval #: 21-507 MR 2111) granted approval to carry out the study within its facilities according to the latest revisions of the Declaration of Helsinki.

## Data Availability

The raw data are available in the Supplemental File.

## Supplemental Information

Supplemental information for this article can be found online at http://dx.doi.org/10.7717/peerj.18163#supplemental-information.

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
