# Peer review of "The effects of resistance training to near volitional failure on motor unit recruitment during neuromuscular fatigue"

_PeerJ, doi:10.7717/peerj.18163_

## Round 0.1 · original submission · Major Revisions

Please find the comments from the reviewers below.

Reviewer 1 ·

Basic reporting

In the present manuscript, during a 5-week intervention, 19 trained volunteers were assigned to two different groups to compare the effect of two different approaches to resistance training (closer or farther away to the volitional failure) on maximal isometric torque, time to exhaustion at a submaximal task, and the evolution of different torque and electromyographical amplitude-related signals. I read the study with interest, and I found several concerns that require attention before publication.

Experimental design

There are two important methodological limitations to the present work:
- The authors prescribed increasing number of %1RM as the protocol progressed through the 5 weeks. The last weeks presented very high % of 1RM (85% and 95%). Especially at 95% load, the difference between low- and high- RIR is probably extremely low, which could explain some of the results.
- Performing the test at the same absolute intensity (i.e. 30% of the MVIC pre intervention) might influence some results regarding fatigability (time to exhaustion).

Thus, the study design does not properly evaluate the effect of low vs high RIR, as this would require at a standardization of the external load between the two modalities, which is quite challenging.
It rather evaluates the non-superiority of a progression from 65% to 95%1RM in training at low repetitions vs high repetition (closer to exhaustion) on some neuromuscular adaptations. This is a crucial point that needs to be clarified in the manuscript.

Validity of the findings

Despite these limitations, I find interesting how performing less effort (lower number of repetitions across the first weeks) lead to similar results in terms of increase in maximal isometric torque, fatigability and other neuromuscular adaptations linked to MU recruitment in the VL. Did the authors collect performance data across the weeks? Assuming the execution quality did not improve, these data would be useful in understanding if the improvements observed in the present manuscript are progressive across the training period or skewed through the last weeks.

Additional comments

Please find here below my specific comments for each section, I hope they would help improving the manuscript.

INTRODUCTION

I find the introduction lacking on some important details regarding the approach proposed (RIR). It would be useful to add some information on how these interventions have been previously performed: a low RIR of 0-1 could be 15/16 (thus relatively low absolute load) or 3/4. From a neuromuscular point of view, the two are different. I suggest the authors to add some details that could help the reader to understand the context (e.g. absolute number of repetitions targeted in previous studies, the absolute volume etc.).

At lines 74 – 78, The authors hypothesized similarities between groups (line 109). Why here the authors speculate on possible different behaviour on sets performed at high vs low RIR? Would not induce different adaptations? Or is it relative to an hypothetical training performed at an hypothetical similar volume?

L74-78 Is it not the opposite? Apologies if I am missing something but for matched volume (e.g. area under the curve, which you did not perform in the present study) low RIR corresponds to an absolute greater load / weight on the bar (or greater time under tension, assuming the range of motion is similar).
If similar volume is reached using a different number of series between low and high RIR, I do not think the possible explanation regarding higher or lower motor drive presented here is appropriate. It would be useful for a potential reader to be more explicit on this point.

What is the rationale behind performing the task at low intensity (30% MVIC)?

METHODS

There is no group control, thus it is not an RCT in the classical definition. It appears that the authors performed a stratified randomization, as participants were allocated according to sex and Wilks score, which makes sense considering the low numerosity and the possible influence of these factors on the outcome. Please modify the manuscript accordingly.

Article 35 of the last declaration of Helsinki states that “every research study involving human subjects must be registered in a publicly accessible database before recruitment of the first subject”. Please provide a link to the registration.

L163 pre-test indicates before intervention and post-test indicates post intervention? Why did the authors choose to adopt the same absolute load? Please provide details about the rationale as, although it would not influence group difference, this could influence the time effect (pre vs post intervention) via different recruitment behaviour (e.g. Dideriksen, Enoka and Farina, 2011) and inflate the result on time to exhaustion.

According to the hypotheses, for the electromyographical data, the authors studied the non-superiority / equivalence of performing a high number of repetition vs lower number of repetitions on the same exercise (similar %1RM and number of sets, supposedly similar time under tension and recovery). I therefore suggest the authors to apply the appropriate statistical approach to test equivalence (an useful resource could be : https://doi.org/10.1113/EP090171).

Why the author chose to analyze only 3 contractions? Analyzing the whole set of contraction fitting a linear model could provide more information. The author computed a similar procedure in the paragraph that follows however only the slope was compared across participants using ANOVAs.

It could be interesting and appropriate to adjust the analyses for sex.


RESULTS

L285-287 Apologies I misunderstood, but if collapsing across time means deleting the time effect and having two time points for each participant, it is not appropriate as it would artificially double the sample-size when you could simply start comparing groups a baseline. I have the same comment for the other successive sections. I suggest writing a paragraph where you described the characteristics and performances of the groups at baseline.

As training progresses, the difference between low to high RIR in term of number of repetitions at a given load probably decreases. Presenting the evolution of RIR (as number) and thus workload (at least external workload) between groups across the 5 weeks is important. However, these data are not presented in the paper. I strongly suggest the authors to present a plot with one line per group, with on the Y axis the RIR number performed and on the X axis the % of 1RM. Finally, a description of the variables presented in supplemental materials would be useful, as I found myself surprised or confused by some torque values presented. I also invite the authors to present the unit of measure and avoid the use of lb or other units that do not belong to the SI. I also suggest the author to present consistently mean ± standard deviation, instead of standard error, so to have a measure of variation.

DISCUSSION

Please amend the discussion according to my main comments and the comments specific to the other sections.

MINOR COMMENTS
INTRODUCTION

L56-58 you provide clear information about the intensity. Please provide a statement regarding volume (e.g. if it was similar between high and low RIR)

L68-74 the information presented here are correct, but the context as presented might not be appropriate. The studies mentioned did not deal directly with the problem of low or high RIR, but rather on neuromuscular responses to fatiguing exercise. I suggest making some changes in the organization of this part to clarify.

METHODS

L132 it might be necessary to provide a reference to normative data. For example, it might help presenting at which percentile of the IPF federation athletes the subjects would fall into (data available at https://www.openpowerlifting.org/rankings/ipf).

L209-215 I could suggest simplifying stating that MUAP amplitude was measured the first, middle and last contraction of the protocol.

Reviewer 2 ·

Basic reporting

In general, the introduction is concise and leads to a clear research question.
Rework of the introduction (using less complex grammer, explaining more the underlying mechanisms of some claims) could benefit the clarity and fluidity of the text, specific suggestions, see attachment.

The methods are detailed, however lengthy, a figure illustrating the (entire) protocol might help the reader to quickly understand the protocol.

For results, raw data are available in the supplementary materials. Statistics are both written and in a table which is redundant. Please consider removing either the table or referring in the text to the table. Furthermore, statistical information is given, but actual values are missing. Table 2 and 4 provide pooled data per main effet which is not sufficient for the reader to understand the aim of your study: how do both groups differ following the training study. I invite the authors to provide detailed data per subgroup instead of pooled dara. More figures on the data instead of tables could also help visualise your results to the reader.

Experimental design

The experimental design is interesting and well explained. Although the design has several nice elements there are also quite some weaknesses aside from those mentioned by the authors (no supervision, only 5 week training period, aspecifity of test vs training).

Anthropometrics are described for males and females. However, since there are two groups being compared, not sex differences, I kindly advise the authors to provide information (and statistical confirmation) whether groups are similar at baseline. The baseline time to task failure visually seems different at baseline between groups. This might affect interpretation: 43s increse starting from 110-115 seconds or from 160s is not necessarily a similar increase from a performance point of view.

Have you considered normalizing the EMG signal for the beginning, middle and end phase to the value at MVC? And if yes, why did you chose not to repor them?

Validity of the findings

The data are interesting, showing no differences in time to task failure and motor unit control in isometric submaximal contractions following exercise training close or less close to volitional exhaustion. This is a very specific finding which is useful in support of, and in a broader context with maximal strength, hypertrophy, but also dynamic exercise.

These aspects are touched upon in the discussion although I feel that the integration/link between all variables could be improved by reworking the discussion a bit to present a more clear message to the reader. A more elaborate 'translation' of what your results (might) mean for resistance training practice, could further help position your findings in this regard.

Statistical details are provided in detail, actual results however, are only mentioned as pooled averages. As such, a lot of information useful for answering the research question is lost (see above).
On a statistical side, the three way ANOVA has 2x2x3 levels, which might be a lot and might complicate interpretation. The lack of significant interaction effects facilitate the interpretation. However, I invite the authors to consider whether the use of deltas (pre - post) to eliminate the factor time and have a more simple statiscal model might help them answer their research question. Also, consider renaming the factor 'Time' to avoid confusion between Time and Time Interval (maybe 'Training'?).

Effect sizes are defined in the statistical section, it would be interesting to use these definitions in the discussion.

Do I understand correctly that you used
A mixed model two way anova for some parameters
A three way 'classic' repeated measures anova for the other?
In that case, was normality checked where needed?

Given the limitations stated in line 434 -434 on the duration (5 weeks) and line 438-441 on the nonspecific test, I encourage the authors to be more careful in their phrasing of the conclusion (line 459-469). With a longer training period, or a test which resembles more the actual movement, maybe you would find other results.

Additional comments

Your refer to the study of Ruple 2023. It looks like this is the same population with the same study design. Why did you not choose separate these data from the Ruple 2023 study instead of combining them to have a more holistic approach on the topic?

Annotated reviews are not available for download in order to protect the identity of reviewers who chose to remain anonymous.

Reviewer 3 ·

Basic reporting

Dear authors, I have carefully revised your original paper entitled “The effects of resistance training to near volitional failure on motor unit recruitment during neuromuscular fatigue”. Data presented herein are a component of a larger study, with the first manuscript having already been published (PMID: 37144554). In that study, they report NO differences in hypertrophic adaptations and improvements in strength but differences in motor unit control during high force contractions between low versus high RIR (0-1 vs 4-6 RIR)training (slopes and intercept of the motor unit mean firing). In this second study, the authors aim to detect differences in MUAP and sEMG during fatiguing contraction along with peak torque and time to exhaustion.
The authors' study on RIR training is an addition to the growing literature in recent years. Their research question, which has not been previously explored, adds a novel perspective to the field.

Overall, the article is well-written and presented. Moreover, the introduction section, experimental procedure, data collection, and analysis have been described rigorously, and the authors tell a clear story.
However, I suggest that you improve the description in lines 333-364 of the discussion.
Lastly, tables 1-4 need to be revised as some data doesn’t agree with what is written in the text, and some repeat what is written. Even if I understand that the number of data is large, I suggest using figures/tables (with just symbols for the significant stat) with the absolute data to show trends/improve the article.
Figures 1 -2 need some captions to describe what they represent

Experimental design

Experimental procedure, data collection, and analysis have been performed rigorously. However, a better description of the resistance training is needed.

Validity of the findings

Despite the multiple limitations mentioned by the authors, especially considering statistical power, the small sample, and the training duration and control. The findings and conclusion are supported by the results and clearly stated.

Additional comments

I have comments for the authors to consider and highlight some aspects of the paper that warrant clarification:

Abstract
Line 28: "very", quite colloquial
Results: 43: add “%” of decrease and increase in both groups.
Line 82: add if volume matched training or not.
Line 98: “discussionS”
Line 94 – 96: not necessary.
Be consistent with resistance-trained or resistance trained among the article.
Line 131-133: add references that justify these subjects as a resistance-trained one based on the squat results that you obtained.

Line 146-153: add references for the MVC protocol
Line 157: why 33 seconds?
Line 164-166: Confused and extensively detailed. I would use the terms pre-intervention and post-intervention instead of pretest and posttest and avoid the example.
--
Resistance Training Program section:
Please add some details on the training program. Moreover, How did you test the 1RM prior to determining the resistance training program?
If the 1RM was correctly determined, in the low RIR, subjects could perform all the predetermined intensity, i.e., 65% to 95% 1RM across the 5 training weeks, going from 15 to 2 reps X set.
But
How did you manage this in the high RIR (4-6)? subjects should be able to perform no more than 2-3 reps (with RIR 4) during the third week at 85%. https://doi.org/10.1007/s40279-023-01937-7
How did you manage the last 2 weeks by adjusting loads and reps?

In the training program, sets for both the Low and high RIR programs are the same, going from 3 to 6 across weeks. I expected different training volumes (set x reps) between the two programs (during the first and second weeks of about 30 to 50%). Can these alter the results (by decreasing training adaptation on the High RIR)?

--
RESULTS section
Line 286-288: add result statistics.
The high RIR group showed 30% higher time-to-task failure on the pre-intervention compared to the low RIR group, and as the authors wrote, if you collapsed across time, that became 25%. The high RIR group is able to perform better from the beginning, reducing the potential for further improvements in this group.

Line 291 -330: Considering collapsing the initial sentences of all these three variables for clarity. For example:
“The results from the three-way mixed factorial ANOVAs (group x time x Time interval) indicated that there were no significant interactions with small to medium effect size (F ≤ 1.899, p ≥ 0.177, ηp2 ≤ 0.100 for all comparisons).”
Line 297; 307 and 318: When you described the “group x time collapsed section,” add p value and effect size after significantly greater or smaller.
Line 305: based on the table I think the effect size should be below < .065
Line 307-311: Just say the “sEMG excitation was smaller during the beginning of the isometric fatigue task “
Line 315: based on the table, I think F should be below < .740
Line 318- 322: Just say: “was significantly greater at the end of the isometric fatigue task (beginning…).”

--
Tables and figures:
Table 1
It should show me something different from the text, not the same data. Moreover, it is confusing, and the main effects of time and group on torque steadiness are not the same as what is written in the text. There is an asterisk on the F for time x time interval.
Same for table 3
Tables 2 and 4 check the overall tables (for example, use 2 decimals in all the described data) and check line space: “group” and “time” are in higher positions than “time x interval.”
Describe absolute data using tables 2 and 4 (or create figures, preferable) and add significant statistics using symbols. Delete tables 1 and 3.

Figure 1:
Add a caption to describe the overall figure

Figure 2:
Add a caption to describe the overall figure

--
Discussion:
Line 334 – 360: It is hard to follow. Please rephrase.
The authors should first describe what they tested, hypothesis and aim and the general findings of their study without using statistics terms like “time point x group interactions”. Here, use near-failure or non-failure training, like in line 347.

Line 347: It's not clear. Do you mean “…..that reported improvement in muscular strength and work capacity by adopting both near failure and non-failure training with no differences between them” ?
Please rephrase
Moreover
“Our findings corroborate previous investigations in muscular strength and work capacity.” Even if you use different variables, I would not recommend an article from the same data collection to confirm the findings (Ruple et al., 2023).

Line 350 seems to be a repetition of line 347.
Line 356 -360 should be moved to line 349 at the end of the sentences to describe muscular strength first. Then, you can continue with work capacity, time-to-task failure, and anaerobic capacity. Be consistent with the order in which you mentioned the variables and with the terms used.

Line 351: compare the improvements in your study compared to Izquierdo (use %)
Line 371: Say of about (in %)
Line 385: between “training groups”;
Line 406: dependence

---

## Round 0.2 · Minor Revisions

Dear authors,

Please find the comments from the reviewers below. I think the comments regarding the study's design and the consistency in terminology could be particularly interesting to implement. If possible, also try to not refer too much to your other article for the methods section (e.g., comment n2 of reviewer 2 in the previous round), this manuscript should be readable standalone.

Reviewer 1 ·

Basic reporting

Thank you for answering my comments. After carefully considering your answer to my question, it appears important to make even more explicit that this represents a secondary analysis, and limit the description of the methodology used even further. The authors already worked on this point, but I think it needs one further revision.
In particular, in the methods there are at times some detailed description of the protocol, so the reader (and myself) expected results and discussion to be presented accordingly (hence my comments #1, #3 etc.). In a nutshell, I can notice that most of the answer to these comments I made were that the information is presented in the main manuscript.
As this is a separate manuscript, the reader should be able to read the manuscript fully without the need to jump from one manuscript to the other to obtain answers to questions that might arise. The authors replied to several methodological concern I raised, and I will not insist on the discussion about the statistical approach adopted. I hope my present comments could help the authors to finalize their manuscript.

Experimental design

The answer to comment #7 should be added to the manuscript in the section isometric torque testing (which I advise to change the name to “isometric fatiguing task” or something similar, as the main aspect of the manuscript is studying the response to fatigue). Please add to the manuscript also the answer to comment #10.

Validity of the findings

In the introduction, I find the rationale behind investigating the activation in fatiguing condition not well presented. This is the main focus of your present work and deserves more in-depth information. Until line 96, the introduction could be shortened, leaving more space to discuss previous findings and justify the hypothesis presented in the present work according to previous findings on isometric fatiguing tasks.

The discussion should be organized better. In particular, at times it is hard to understand if the authors are discussing the effect of training in absence or presence of fatigue. I would recommend focusing more on the effect of training plus presence of fatigue.

Finally, I advise to clarify explicitly somewhere in the discussion the arguments made in comment #1 of the previous round, despite all the efforts made by the author to clarify the protocol, one could still argue that the loads in the last phases of the training, being extremely high, made the low and high RIR trainings looking similar, masking some possible differences. Comment #1 is maybe one of the most important points to clarify in the manuscript.

Additional comments

77-79 This is probably a follow up of comment 6 in the previous round: this part remains confusing. It is probably true for lower loads. For high workloads it is less likely to observe this pattern, since high-treshold motor units would be recruited to sustain the high load.


362 please reconsider in several sentence the use of “similar” (sometimes it is possible to replace with “not different” etc). To me and possibly to many readers, “similar” would possibly imply that a test for similarity or equivalence was performed (as discussed in the previous round).

369-371 near failure, non-failure, non-muscle failure… some of them are the same as low and high RIR? Please be consistent in the terminology used to avoid confusion.

393-408 this paragraphs could be simplified to one sentence (i.e. low or high RIR groups presented similar decrease in force steadiness), as in the current version, it is more presented the effect of fatigue on force steadiness (already described in several populations in the literature) and the physiological mechanism behind it.

410-432 here I’d advice to develop the reasoning of line 77-79: what would change between low and high RIR, how the evidence from the literature could explain the current results? I find it a big limitation to present separately the physiology and the results of the present study.

434-454 Please put forward results from the present study.

Reviewer 3 ·

Basic reporting

The authors responded satisfactorily to all of my previous criticisms. I have no other further questions.

Experimental design

The authors responded satisfactorily and improved the methodological section following the criticism reported.

Validity of the findings

Tables, figures, and discussion sections are now improved and clearer for the reader.

---

## Round 0.3 · accepted · Accept

Thank you for your submission to PeerJ.

I am writing to inform you that your manuscript - The effects of resistance training to near volitional failure on motor unit recruitment during neuromuscular fatigue - has been Accepted for publication. Congratulations!

Reviewer 1 ·

Basic reporting

thank you for answering my questions and comments. I think this work will be of great interests to the readers and a nice add to the present literature.

Experimental design

.

Validity of the findings

.

Additional comments

.